# Usefulness of the Modified Videofluoroscopic Dysphagia Scale in Evaluating Swallowing Function among Patients with Amyotrophic Lateral Sclerosis and Dysphagia

**DOI:** 10.3390/jcm10194300

**Published:** 2021-09-22

**Authors:** Byung Joo Lee, Hyoshin Eo, Donghwi Park

**Affiliations:** 1Department of Rehabilitation Medicine, Daegu Fatima Hospital, Daegu 41133, Korea or bjl84@hanmail.net (B.J.L.); wowo0101@naver.com (H.E.); 2Department of Physical Medicine and Rehabilitation, College of Medicine, Ulsan University Hospital, University of Ulsan, Ulsan 44033, Korea

**Keywords:** modified videofluoroscopic dysphagia scale, deglutition, amyotrophic lateral sclerosis, videofluoroscopic dysphagia scale

## Abstract

Introduction: The videofluoroscopic dysphagia scale (VDS) is used to predict the long-term prognosis of dysphagia among patients with the condition. Previously, a modified version of the VDS (mVDS) was established to overcome the relatively low inter-rater reliability of VDS, and was verified in patients with dysphagia, such as stroke patients. However, the validity of mVDS in patients with amyotrophic lateral sclerosis (ALS) has never been proved. Therefore, in this study, we attempted to seek the validity of the mVDS score in patients with ALS suffering from dysphagia. Method: Data from the videofluoroscopic swallowing study (VFSS) of 34 patients with ALS and dysphagia were retrospectively collected. We investigated the presence of aspiration pneumonia and the selected feeding method based on the VFSS. We also evaluated the correlations between the mVDS and the selected feeding method, and between the mVDS and the presence of aspiration pneumonia. Multivariate logistic regression and receiver operating characteristic (ROC) analyses were performed during the data analysis. Results: In patients with ALS and dysphagia, the mVDS scores were statistically correlated with the selected feeding method (*p* < 0.05) and the presence of aspiration pneumonia (*p* < 0.05). In the ROC curve analysis, the area under the ROC curve values for the selected feeding method and the presence of aspiration pneumonia were 0.886 (95% confidence interval (CI), 0.730–0.969; *p* < 0.0001) and 0.886 (95% CI, 0.730–0.969; *p* < 0.0001), respectively. Conclusion: The mVDS can be a useful tool for quantifying the severity of dysphagia and interpreting the VFSS findings in patients with ALS and dysphagia. However, further studies involving a more general population of patients with ALS are needed to elucidate a more accurate cut-off value for the allowance of oral feeding and the presence of aspiration pneumonia.

## 1. Introduction

Amyotrophic lateral sclerosis (ALS) is a progressive and fatal neurodegenerative disease affecting both the upper and lower motor neurons within the cortex, the brainstem, and the spinal cord [1,2,3,4]. Dysphagia, or swallowing impairment, reportedly occurs in 85% of patients with ALS at some point during the disease process, and is associated with malnutrition, weight loss, reduced quality of life, aspiration pneumonia, and death [5]. In patients with ALS, early detection and consistent monitoring of dysphagia provide the opportunity to mitigate the associated risks and improve survival through timely interventions [6]. Therefore, dysphagia evaluation in patients with ALS is necessary for detecting bulbar involvement as well as improving the survival rate. 

Among various methods to evaluate dysphagia, the videofluoroscopic swallowing study (VFSS) and the fiberoptic endoscopic evaluation of swallowing (FEES) are the gold standard [7]. The FEES and VFSS are complementary procedures. VFSS can be used to evaluate problems in the oral phase of the swallowing process. In addition, intradeglutitive silent aspiration can be detected. The FEES, in the other hand, can be used to directly visualize the swallowing process, and saliva movement can be detected. 

The interpretation of VFSS results requires deep knowledge of the swallowing process and a lot of experience in monitoring the VFSS [7,8]. The videofluoroscopic dysphagia scale (VDS) is a based on the VFSS results and is used to anticipate the prognosis of dysphagia in patients with swallowing difficulty [9]. The VDS contains 14 categories and is well correlated with aspiration or penetration symptoms that occur after dysphagia [9]. The 14 categories represent both oral functions (lip closure, mastication, bolus formation, premature bolus loss, apraxia, and oral transit time) and pharyngeal functions (pharyngeal triggering, laryngeal elevation, epiglottic closure, pharyngeal transit time, pharyngeal coating, vallecular and pyriform sinus residues, and tracheal aspiration) [9].

The VDS can quantify the severity of dysphagia through a total score. However, issues regarding VDS scoring have been noted in previous studies. Kim et al. reported that the inter-rater reliability of VDS had a low rate of agreement (κ < 0.20) in the bolus formation (κ = 0.153), mastication (κ = 0.123), apraxia (κ = 0.099), tongue-to-palate contact (κ = 0.153), premature bolus loss (κ = 0.060), and pharyngeal transit time (κ = 0.165) categories, despite other categories exhibiting a fair rate of agreement (κ > 0.2, κ < 0.4) [10]. Such outcomes mean that the VDS could be subjective depending on the interpreters, especially in several categories such as apraxia, tongue-to-palate contact, premature bolus loss, or bolus formation [10]. This relatively low inter-rater reliability of some categories in the VDS could be a result of an ambiguous criterion of corresponding parameters [10].

A modified version of VDS (mVDS) has been developed to overcome such problems [11,12]. A previous study revealed that a modification in some categories of VDS enhanced the inter-rater reliability of the evaluation [11]. Moreover, it was well correlated with the feeding method selection and the presence of aspiration pneumonia. However, these previous studies were only performed in patients who suffered from stroke and etiologies other than ALS. Therefore, in this study, we attempted to evaluate the validity of the mVDS score in patients with ALS and dysphagia.

## 2. Methods

### 2.1. Ethics Statements

The protocol for this study was approved by the Institutional Review Board of Ulsan University Hospital (2021-01-028). This study was conducted according to the Declaration of Helsinki for human experiments. 

### 2.2. Participants

We retrospectively reviewed the medical records of patients who visited our ALS clinic between January 2014 and August 2020. ALS diagnosis was established based on the revised El Escorial criteria [13]. Thirty-four patients with clinically definite, probable, or probable laboratory test-supported ALS were included in our study. We recorded age, sex, region of initial symptom presentation, total ALS functional rating scale—revised (ALSFRS-R) scores, feeding method based on VFSS, mVDS score, and scores on the penetration–aspiration scale (PAS). The exclusion criteria were (1) patients with ALS who had not undergone VFSS; and (2) the presence of other previously diagnosed diseases that could cause dysphagia such as stroke, Parkinson’s disease, or laryngeal cancer.

### 2.3. The VFSS Protocol

The VFSS was performed with a fluoroscopic device and was recorded as a video file. During the VFSS, patients consecutively swallowed the following materials that had a stepwise increase in consistency: water, nectar (51–350 cP), rice porridge (351–1,750 cP), and boiled rice (>1,750 cP) [14]. The materials were mixed with liquid barium, and the patient swallowed them while in a relaxed sitting position. Dynamic fluoroscopic images were obtained in the anterior–posterior and lateral views and were recorded at 30 frames per second. The VFSS images were analyzed according to the PAS and were considered positive for aspiration if the PAS score was >5 [15].

All studies were reviewed by 2 physiatrists who had at least 7 years of experience in interpreting VFSS results. Patient information, including age, sex, and underlying diseases, was withheld from the interpreters. The interpreters only observed the patients using the video files on their laptops, described their findings, and chose a feeding method (non-oral feeding versus oral feeding) based on the VFSS results.

### 2.4. Modification of the VDS

The mVDS was developed based on a study regarding the inter-rater reliability of the VDS. Among the VDS categories, the ones with a κ value < 0.2 (bolus formation, mastication, apraxia, tongue in palate contact, and pharyngeal transit time) were modified (Table 1) [11,12]. As mentioned by previous researchers, such categories have somewhat ambiguous guidelines and 3 to 4 multiple options, which lead to low reliability. Therefore, the mVDS was developed by modifying the mentioned categories to a binary scale or deleting the ambiguous categories. The mVDS with a sum of 100 points was created according to the odds ratios of various prognostic factors to measure these VFSS findings as objective quantitative scores (Table 2) [11,12].

### 2.5. Aspiration Pneumonia

A retrospective review was conducted among patients with ALS and dysphagia to investigate the development of aspiration pneumonia within a month before and after a VFSS examination. The following data were collected: respiratory symptoms, such as coughing during feeding; the presence of sputum, dyspnea, or fever; chest X-ray findings; blood laboratory findings (white blood cell (WBC) counts, C-reactive protein (CRP) level, and erythrocyte sedimentation rate (ESR)); and use of antibiotics [16,17,18].

Although arriving at a definitive diagnosis of aspiration is difficult and despite the diagnostic criteria for aspiration pneumonia being slightly different across various studies, patients who met all of the following criteria were considered to have aspiration pneumonia in the present study: (1) the presence of both objective signs (coarse lung sounds, the presence of lung infiltration on chest X-ray, and systemic inflammation based on blood laboratory findings such as increased CRP levels and WBC counts) and subjective symptoms (fever, cough, and increased purulent sputum); (2) clinical suspicion of aspiration (delayed swallowing or coughing during swallowing); and (3) no evidence of microorganisms such as Legionella or Mycoplasma, which are common pathogens in atypical pneumonia. In addition, the clinical reports from the internal medicine department were used to confirm the diagnosis for aspiration pneumonia [16,17,18].

### 2.6. Statistical Analyses

To evaluate the correlation between the mVDS and the selected feeding method as well as between the mVDS and the presence of aspiration pneumonia in patients with ALS, univariate logistic regression analysis with the enter method was used. To evaluate the accuracy of predictive factors for the presence of pneumonia and allowance of oral feeding based on the VFSS findings, we performed receiver operating characteristic (ROC) analysis. Statistical analyses were conducted using the MedCalc program (MedCalc Software, Ostend, Belgium) and SPSS software version 22.0 (IBM Corp., Armonk, NY, USA).

## 3. Results

### 3.1. Patient Characteristics

A total of 34 patients with ALS (20 males and 14 females) were included in this study. Among them, 12 patients had limb-onset ALS and 22 had bulbar-onset ALS. Their mean ALSFRS-R score was 29.82 ± 8.47. In total, 20 patients were fed orally, whereas 14 patients used a Levin tube. A total of 29 patients were diagnosed with aspiration pneumonia and had received antibiotics. 

### 3.2. Inter-Rater Reliability of the mVDS

The inter-rater reliability (Cronbach α value) of the total mVDS score was 0.886, which was consistent with very good inter-rater reliability.

### 3.3. Relationship between mVDS and Feeding Method

In the univariate logistic analysis, the mVDS score was statistically correlated with the selected feeding method (*p* < 0.05) In the ROC curve analysis, the area under the ROC curve (AUC) for the selected feeding method was 0.886 (95% confidence interval (CI), 0.730–0.969; *p* < 0.0001). The optimal cut-off value for the oral feeding obtained from the maximal Youden index was a score of >69.5 based on the mVDS (sensitivity, 71.43%; specificity, 100.0%) for non-oral feeding (Figure 1A, Table 3). 

### 3.4. Relationship between mVDS and Aspiration Pneumonia

In the univariate logistic analysis, the mVDS score was statistically correlated with the presence of aspiration pneumonia (*p* < 0.05) In the ROC curve analysis, the AUC for the selected feeding method was 0.886 (95% CI, 0.730–0.969; *p* < 0.0001). The optimal cut-off value for the development of aspiration pneumonia obtained from the maximal Youden index was a score of >87 based on the mVDS (sensitivity, 80.0%; specificity, 93.1%) (Figure 1B, Table 4).

## 4. Discussion

No previous study has investigated the validity and usefulness of mVDS in evaluating dysphagia among patients with ALS. In the present study, the mVDS score showed significant correlations with the presence of aspiration pneumonia and the feeding method based on the VFSS interpretation by physiatrists among patients with ALS and swallowing difficulty. 

Like the VDS, a higher mVDS score indicates a greater need for diet limitation and more severe dysphagia. The result of this study suggested that an mVDS score ≥87 was significantly correlated with the incidence of aspiration pneumonia (sensitivity, 80.0%; specificity, 93.1%). Moreover, an mVDS score ≥69.5 was significantly correlated with no allowance of oral feeding (sensitivity, 71.43%; specificity, 100.0%). In previous study which evaluated the usefulness of mVDS in stroke patients with dysphagia, the optimal cut-off value obtained from the maximal Youden index for non-oral feeding was a score of ≥36.5 based on the mVDS (sensitivity, 92.86%; specificity, 76.19%). Additionally, a score of ≥67 showed a sensitivity of 28.57% and specificity of 100% for non-oral feeding. Although there was a difference between the results of this study and the previous study based on the optimal cut-off value of non-oral feeding (36.5 vs. 69.5), similar results were obtained when the specificity was 100% (69.5 vs. 67). However, to obtain more accurate results, additional studies with a larger number of patients with various diseases will be needed.

There are numerous methods to evaluate dysphagia. The simplest method would be a bedside swallowing examination, wherein patients are asked to perform a swallowing motion and the physicians simultaneously palpate the neck area to examine the hyoid motions according to the swallowing process [19,20]. This method is simple and easy to apply but reportedly has far from high sensitivity and specificity. Another method is the fiberoptic endoscopic evaluation of swallowing, wherein a fiberoptic endoscope is inserted through the nostril and into the throat to directly visualize structures and the oral–pharyngeal transfer of swallowed material [21,22]. However, patients often complain of nasal discomfort, and in some patients with anatomical variations it would be very difficult for the endoscope to pass through the nasal cavity. In addition, this evaluation cannot detect problems present in the oral cavity or esophagus [21,22]. By contrast, the VFSS can overcome the limitations of these two examinations. The VFSS does not involve the insertion of foreign objects in the body, and it can detect problems in any stage of the swallowing process, such as in the oral, pharyngeal, and esophageal phases [23,24,25]. However, radiation exposure is a concern for VFSS [26]. Moreover, the interpretation of VFSS results is somewhat complicated and requires skills and experience in the field. Nonetheless, the VFSS can visualize the entire swallowing process, from the oral phase to the esophageal phase, and is the gold standard in dysphagia evaluation [7].

Numerous existing methods can be implemented in interpreting the VFSS results using the VFSS video to quantify the severity of dysphagia. One of them is the PAS, which is an eight-point scale used to describe the depth of invasion and response to airway infiltration during VFSS [27]. It can define the extent of airway infiltration well; however, the correlation between PAS score and feeding method based on VFSS results is reportedly weak [9,12]. Another method for interpreting VFSS is the VDS [10]. As previously mentioned, the VDS is a 14-item tool with weighted values that exhibit oral and pharyngeal functions observed in the VFSS. Previous studies have shown that it is well correlated with aspiration and/or penetration occurring 6 months after the initial onset of dysphagia [10]. The VDS can also denote the severity of dysphagia in quantifiable scores [9,10]. However, due to the complexity and ambiguity of some VDS categories, the VDS has been reported to have low inter-rater reliability [10]. To supplement these disadvantages, some categories of VDS have been modified, and its application to patients with stroke or other etiologies except ALS who are also suffering from dysphagia was proven valid in previous studies. Similar to the results of these studies, our results suggest that the mVDS is a useful tool for quantifying the severity of dysphagia and for interpreting VFSS findings in patients with ALS and dysphagia. Similar to the VDS, a higher mVDS score indicates greater diet limitation and more severe dysphagia. The mVDS provides numerical data based on swallowing function by using comprehensive VFSS findings with a relatively high inter-rater reliability. Therefore, in patients with ALS, the mVDS provides more intuitive data compared to other conventional VFSS interpretation tools, which are usually focused on the presence of aspiration or penetration.

There are some limitations in this study. The first is the small number of participants, which is challenging when making a general conclusion. A greater number of patients with ALS would be necessary for a more cohesive outcome. Second, the participants were limited to patients with ALS and dysphagia symptoms who had undergone VFSS. Patients with ALS who have no bulbar or dysphagia symptoms could exhibit abnormal VFSS results. To apply the results of this study to the general ALS population, further studies that include patients with ALS who do not have dysphagia symptoms are necessary. Third, the results could have been different when performing an analysis of in limb-onset ALS and bulbar-onset ALS, separately. In addition, the results could be interesting when performing a further analysis between ALS patients with and without aspiration pneumonia, or between ALS patients with and without L-tubes. In this study, however, the number of patients was not large enough to extract meaningful results. A greater number of patients with ALS would be necessary for a more cohesive outcome. Fourth, the mVDS may not properly reflect the dysphagia of ALS patients, as most dysphagia in ALS patients occurs in oral phase. Among nine parameters of mVDS, three are from the oral phase. For example, with a score of 21.5 out of 100, 21.5% is from the oral phase. Considering that the most general aspiration occurs during the pharyngeal phase, this score portion does not seem insignificant.

## 5. Conclusions

The mVDS can be a useful tool for quantifying the severity of dysphagia as well as in interpreting the VFSS findings in patients with ALS and dysphagia. However, further studies involving a more general population of patients with ALS are needed to elucidate a more accurate cut-off value for the allowance of oral feeding and the presence of aspiration pneumonia. 

## Figures and Tables

**Figure 1 jcm-10-04300-f001:**
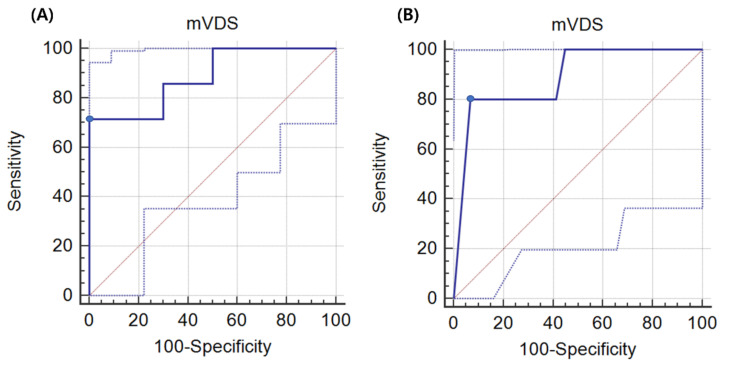
(**A**) ROC curve of the mVDS score for the selection of oral feeding in patients with ALS and dysphagia. The optimal cut-off value (dots on the curves) of the mVDS score, which was obtained from the maximal Youden index, was >69.5 (AUC, 0.886; 95% CI, 0.730–0.969; *p* < 0.0001; sensitivity, 71.43%; specificity, 100.0%). (**B**) ROC curve of the mVDS score for the development of aspiration pneumonia in patients with ALS and dysphagia. The optimal cut-off value obtained from the maximal Youden index was a score of >87 based on the mVDS (AUC, 0.886; 95% CI, 0.730–0.969; *p* < 0.0001; sensitivity, 80.0%; specificity, 93.1%) for the selection of oral feeding. ROC: receiver operating characteristic, AUC: area under the receiver operating characteristic curve, CI: confidence interval, mVDS: modified version of the videofluoroscopic dysphagia scale.

**Table 1 jcm-10-04300-t001:** Modified version of the Videofluoroscopic Dysphagia Scale.

Parameters		Score
lip closure	intact/not intact	0/6
massification	possible/not possible	0/11.5
oral transit time	≤1.5 s/>1.5 s	0/4
triggering pharyngeal swallow (swallowing reflex)	intact/delayed	0/7
epiglottis inversion	yes/no	0/13
valleculae residue	0%/<10%/≥10%, <50%/≥50%	0/3/6/9
pyriformis residue	0%/<10%/≥10%, <50%/≥50%	0/6.5/13/19.5
pharyngeal wall coating	no/yes	0/13
aspiration	intact/penetration/aspiration	0/8.5/17
total score		100

**Table 2 jcm-10-04300-t002:** Characteristics of ALS patients with dysphagia in the present study.

Characteristics	Mean ± Standard Deviation (Minimum–Maximum)
Age (year)	68.53 ± 11.10 (49–82)
Sex (male:female)	20 (58.8%):14 (11.2%)
Duration of disease (month)	25.76 ± 16.39 (2–64)
Limb onset:Bulbar onset	12 (35.3%):22 (64.7%)
Total ALSFRS-R score	29.82 ± 8.47 (10–42)
Tracheal tube (yes:no)	30 (88.2%):4 (11.8%)
Oral feeding:Levin tube feeding	20 (58.8%):14 (11.2%)
History of aspiration pneumonia (yes:no)	5 (14.7%):29 (85.3%)
PAS grade	5.06 ± 2.64 (1–8)
mVDS scores	
Lip closure	1.76 ± 2.78 (0–6)
Mastication	3.38 ± 5.32 (0–11.5)
Oral transit time	3.29 ± 1.55 (0–4)
Triggering of pharyngeal swallowing	6.18 ± 2.29 (0–7)
Epiglottis inversion	3.82 ± 6.01 (0–13)
Valleculae residue	6.18 ± 2.20 (3–9)
Pyriformis residue	12.24 ± 6.35 (0–19.5)
Pharyngeal wall coating	7.65 ± 6.49 (0–13)
Aspiration	11.5 ± 7.21 (0–17)
Total score	56.0 ± 29.15 (9.5–100)

PAS: penetration–aspiration scale, ALSFRS-R: revised amyotrophic lateral sclerosis functional rating scale, mVDS: modified videofluoroscopic dysphaga scale, MMSE: mini-mental status examination, MBI: modified Bathel Index.

**Table 3 jcm-10-04300-t003:** Univariate logistic regression analysis (with the enter method) of the association between the modified version of the Videofluoroscopic Dysphagia Scale scores and the selection of the oral feeding method.

	Parameter	Beta Coefficient	Standard Error	OR (95% CI)	*p*-Value
Selection of oral feeding method	mVDS score	0.078	0.025	1.081 (1.031–1.132)	**0.02**

mVDS, modified version of the Videofluoroscopic Dysphagia Scale; OR, odds ratio; CI, confidence interval.

**Table 4 jcm-10-04300-t004:** Univariate logistic regression analysis (with the enter method) of the association between the modified version of the Videofluoroscopic Dysphagia Scale and the development of aspiration pneumonia.

	Parameter	Beta Coefficient	Standard Error	OR (95% CI)	*p*-Value
Development of aspiration pneumonia	mVDS score	0.070	0.031	1.073 (1.013–1.116)	**<0.001**

mVDS, modified version of the Videofluoroscopic Dysphagia Scale; OR, odds ratio; CI, confidence interval.

## Data Availability

Data available on request due to restrictions eg privacy or ethical. The data presented in this study are available on request from the corresponding author. The data are not publicly available due to privacy reason.

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
