# Peer review of "Usefulness of the Modified Videofluoroscopic Dysphagia Scale in Evaluating Swallowing Function among Patients with Amyotrophic Lateral Sclerosis and Dysphagia"

_jcm, 2021, doi:10.3390/jcm10194300_

Round 1
Reviewer 1 Report
Comments:
Results: How old were the patients?
To show the results for the limb onset and bulbar onset groups separately would be interesting.
discussion:
FEES and VFSS are complementary examination procedures. Both methods are gold standard. In FEES, sensitivity and saliva can be assessed. In VFSS, oral phase and intradeglutitive silent aspiration can be detected.
Author Response
Response to Reviewer`s Comment
Reviewer 1
Results: How old were the patients?
To show the results for the limb onset and bulbar onset groups separately would be interesting.
Answer: Thank you for the comment. We have performed additional analysis following your advice. The number of limb-onset and bulbar onset patients were 12 (35.3%) and 22 (64.7%), respectively. We have added this analysis in table 1. In addition, we have performed separate analysis of each group. Interestingly, mVDS score was only statistically correlated with feeding method in patients with bulbar onset, whereas development of aspiration pneumonia showed no significant correlation with both groups. However, due to the small number of patients with ALS, we did not add the finding in the result section, but mentioned the necessity of such analysis in the further study with large number of ALS patients.
<relationship between mVDS and feeding method in ALS with limb-onset>
< relationship between mVDS and feeding method in ALS with bulbar-onset>
< relationship between mVDS and aspiration pneumonia in ALS with limb-onset>
< relationship between mVDS and aspiration pneumonia in ALS with bulbar-onset>
discussion:
FEES and VFSS are complementary examination procedures. Both methods are gold standard. In FEES, sensitivity and saliva can be assessed. In VFSS, oral phase and intradeglutitive silent aspiration can be detected.
Answer: Thank you for your comment. We have added the information in the manuscript. (introduction part).

Reviewer 2 Report
In this manuscript (jcm-1360910), the authors have analyzed the findings of VFSS of 34 patients with ALS using a modified version of the VDS. They showed optimal cut-off value for the oral feeding and the development of aspiration pneumonia and usefulness of the mVDS. This MS has a special meaning in suggesting that the mVDS can also be applied to ALS patients, but I have some concerns about this MS:
Major comments
- Methods
Comparing with the VDS, the mVDS seems to weigh the pharyngeal stage. However, ALS patients have many problems in the oral stage, poor bolus formation, or transit to the pharynx, because of tongue dysfunction. The authors should present evidence for the advantage of the mVDS over the VDS in evaluating ALS patients.
It is more convenient for readers to add a table about the parameters and score of mVDS.
- Results
Please show the difference in ALSFRS-R and mVDS between limb onset and bulbar onset patients, between oral feeding and tube feeding patients, or between patients with and without development of aspiration pneumonia.
It is better to compare the VDS scores with the mVDS scores to present evidence for the advantage of the mVDS.
- Discussion
Although it is difficult to make a simple comparison between the mVDS and the VDS, the scores of the mVDS, >87 or >69.5, seem to be too high to indicate the development of aspiration pneumonia or allowance of oral feeding. A previous study showed the mean VDS scores of limb onset and bulbar onset ALS patients, 25.0 and 16.2. Please add a discussion about the cut-off values citing this reference.
Umemoto G, Furuya H, Tsuboi Y, et al. Characteristics of tongue and pharyngeal pressure in patients with neuromuscular diseases. Degenerative Neurological and Neuromuscular Disease. 2017
Author Response
Reviewer 2
In this manuscript (jcm-1360910), the authors have analyzed the findings of VFSS of 34 patients with ALS using a modified version of the VDS. They showed optimal cut-off value for the oral feeding and the development of aspiration pneumonia and usefulness of the mVDS. This MS has a special meaning in suggesting that the mVDS can also be applied to ALS patients, but I have some concerns about this MS:
Major comments
Methods
Comparing with the VDS, the mVDS seems to weigh the pharyngeal stage. However, ALS patients have many problems in the oral stage, poor bolus formation, or transit to the pharynx, because of tongue dysfunction. The authors should present evidence for the advantage of the mVDS over the VDS in evaluating ALS patients.
It is more convenient for readers to add a table about the parameters and score of mVDS.
Answer: We agree with your comment and have added additional information in table 1. Among nine parameters of mVDS, three are from oral phase. Score of 21.5 out of 100, in another words, 21.5% is from oral phase. Therefore, the total score of mVDS can reflect both oral phase and pharyngeal phase. Moreover, considering that the most of aspiration is occured during pharyngeal phase, this score portion does not seem insignificant. However, we have added this information in the limitation section.
Results
Please show the difference in ALSFRS-R and mVDS between limb onset and bulbar onset patients, between oral feeding and tube feeding patients, or between patients with and without development of aspiration pneumonia. It is better to compare the VDS scores with the mVDS scores to present evidence for the advantage of the mVDS.
Answer: Thank you for the comment. According your comment, we re-analyzed our data as follows;
< difference in ALSFRS-R and mVDS between limb onset and bulbar onset patients>
Two groups showed no difference.
< difference in ALSFRS-R and mVDS between patients with and without development of aspiration pneumonia.>
The PAS, mVDS total score, and ALSFRS-R score were significantly lower in aspiration pneumonia group.
< difference in ALSFRS-R and mVDS between patients with and without oral feeding.>
The above analysis showed that PAS, mVDS, and ALSFRS-R score were all worse in L-tube feeding group. We will add this information in the limitation section. However, the number of patients were small, we couldn`t add this results in the manuscript. Therefore, we have this limitation in the limitation part.
Discussion
Although it is difficult to make a simple comparison between the mVDS and the VDS, the scores of the mVDS, >87 or >69.5, seem to be too high to indicate the development of aspiration pneumonia or allowance of oral feeding. A previous study showed the mean VDS scores of limb onset and bulbar onset ALS patients, 25.0 and 16.2. Please add a discussion about the cut-off values citing this reference.
Umemoto G, Furuya H, Tsuboi Y, et al. Characteristics of tongue and pharyngeal pressure in patients with neuromuscular diseases. Degenerative Neurological and Neuromuscular Disease. 2017
Answer: Thank you for the comment. We have added above article in the reference. The high score seems to be due to small sample size. The necessicity for larger sample sized study has been added in the discussion section

Round 2
Reviewer 2 Report
In this manuscript (jcm-1360910), the authors revised the first version and added some limitations including further analysis and the small sample size. However, I could not find a new comment in Discussion on the validity of the high cut-off values of mVDS. I am concerned that these high cut-off values may be useless to detect the risk of aspiration pneumonia or tube feeding. Please discuss the cut-off values citing a previous report.
Author Response
Answer: We appreciate your valuable comment. We totally agree with your comment. In previous study, which evaluated the usefulness of mVDS in stroke patients with dysphagia, the optimal cut-off value obtained from the maximal Youden index for non-oral feeding was a score of ≥36.5 based on the mVDS (sensitivity, 92.86%; specificity, 76.19%). Additionally, a score of ≥67 showed a sensitivity of 28.57% and specificity of 100% for non-oral feeding. Although there was a difference between the results of this study and the previous study based on the optimal cut-off value of non-oral feeding (36.5 vs. 69.5), similar results were obtained when the specificity was 100% (69.5 vs. 67). However, to obtain more accurate results, additional studies with a larger number of patients with various diseases will be needed. Therefore, we have added it in discussion part.
